# How Healthcare Teams Appear to the Observer—An Approach to the Complexities of Mask-Wearing Attitudes During the COVID-19 Pandemic

**DOI:** 10.3390/ijerph22111755

**Published:** 2025-11-20

**Authors:** Hans Joachim Roehrens, Michaela Stratmann, Jan P. Ehlers

**Affiliations:** Department of Didactics and Educational Research in Health Science, Faculty of Health, Witten/Herdecke University, 58455 Witten, Germany; michaela.stratmann@uni-wh.de (M.S.); jan.ehlers@uni-wh.de (J.P.E.)

**Keywords:** FFP2 masks, healthcare teams, COVID-19, Health Belief Model, public health, healthcare professionals, mask-wearing, attitudes toward health measures, demographic factors, MANOVA analysis

## Abstract

The COVID-19 pandemic has brought widespread attention to the use of protective masks. This study explores how the socio-demographic factors influence the perception of an interprofessional team wearing FFP2 masks. An online survey conducted in 2021 included 906 participants who rated photographs of an interprofessional team in four different attire scenarios: tunic, white coat, FFP2 masks, and pink socks. In addition education and occupation of participants were obtained. By measuring the attitudes towards an interprofessional team wearing FFP2 masks implicitly the attitude towards this COVID-19 measure was assessed. The statistical analysis employed MANOVA to assess the significance of differences in perception. Healthcare professionals were more critical of teams wearing masks compared to other occupational groups, with a third expressing skepticism about the effectiveness of masks. The results underscore the importance of understanding healthcare professionals’ attitudes towards preventive measures, especially masks, for future pandemics. The study uses the Health Belief Model (HBM) as an approach to interpret how the attitude towards preventive measures during the COVID-19 pandemic affect behavior.

## 1. Introduction

Ever since the beginning of the COVID-19 pandemic, there has been an occasionally vigorous discussion about the wearing of protective masks. This topic was central in the debate about governmental protective measures all over the world [1,2,3,4]. There is scientific and societal consensus that masks are highly valued in protecting against SARS-CoV-2 infection [5,6,7,8]. To achieve the best possible protection of a population during a pandemic, it is of vital importance that individuals demonstrate a profound understanding of how to carry out health measures and thus clearly recognize the importance of their own health behaviors in terms of contributing to a community’s protection and well-being [9].

However, the evidence base for mask-wearing still appears mixed [10]. The fourth update of the Cochrane Review “Physical interventions to interrupt or reduce the spread of respiratory viruses”, published in early 2023, generated extensive debate regarding the protective effects of masks. The review found little or no protective effect for wearing protective masks when it assessed 78 studies with a randomized controlled design [11]. This statement quickly received a high level of attention in the (social) media and led to considerable misinterpretation regarding the usefulness of wearing masks. In response, and in order to pre-empt misinterpretations, Cochrane promptly issued a statement pointing out the limitations of the studies investigated and emphasizing that, ultimately, further studies were urgently needed to further assess the evidence base [12].

Thus, the wearing of masks remains a highly emotional issue that has caused ongoing discussions and conflicts in many societies and countries during the COVID-19 pandemic [13]. Only when the “mask requirement” was abolished in Germany in the spring of 2023 did the turmoil subside. With the onset of each new wave of infections in the future, the mask discussion will certainly become accentuated again, especially in the healthcare sector.

The scientific literature has generated an abundance of studies attempting to classify and understand the attitudes of individuals regarding mask use [14,15,16]. Many facets of human attitudes and actions have been illuminated to date. At the core of the consideration is whether and exactly how human behavior, and particularly health behavior, can be explained, predicted, and modified [17,18,19,20].

There are approximately 6 million employees in the German healthcare system, with 3.6 million having direct patient contact. Their role-model function serves as an important factor in adopting more health-conscious behaviors [21,22,23,24]. In portraying and applying adequate protective measures during the corona pandemic, the behavior of healthcare professionals in general was highly important [16,25]. This will likely also be relevant in future pandemics [26,27].

In order to understand the demand for an individual to engage in health behaviors on a more profound level, we need a model that allows us to understand and predict health behaviors [28,29,30]. In this case, the Health Belief Model contributes a deeper understanding of COVID-19-specific health behaviors [31,32]. The theory states that the likelihood of any given person displaying a particular health behavior depends largely on the experienced threat of the disease [33,34]. Feeling more vulnerable generates a higher experience of fear and losing control. Expectations of protective actions are based on a cost–benefit calculation, weighing perceived benefits against costs. The likelihood of performing a particular behavior results from the evaluation of internalized fear and perceived powerlessness relative to the cost of potential protective measures. Social, psychological, and demographic factors further influence the prediction of health behaviors in terms of the Health Belief Model [35,36,37].

When transferring this theory to the realm of the COVID-19 pandemic, we suggest that the will to apply preventive actions, i.e., mask-wearing, is influenced by perceptions of one’s subjective disease risk [20].

While numerous studies have addressed mask-wearing during the COVID-19 pandemic, almost nothing is known about the professional and educational background influencing the visual perception of healthcare teams wearing masks. The study addresses this gap in terms of combining socio-demographic variables and visual assessment of healthcare teams under the framework of the Health Belief Model (HBM).

In 2021, prior to this study, Roehrens et al. [38] examined the influence of healthcare team attire on patient perceptions, focusing on socio-demographic factors. Participants from diverse socio-demographic backgrounds, including various age groups, genders, and places of residence, were asked to rate healthcare teams on attributes such as sympathy, competence, and trust, based on their attire. The results highlighted the role of socio-demographic factors in determining patient preferences, with implications for tailoring healthcare provider appearance to match the expectations of diverse patient groups. The study presented here was conducted during the COVID-19 pandemic. Interestingly, the team photo which featured all members wearing protective FFP2 masks was perceived as significantly more competent and confident by participants across all age groups. To further understand and evaluate the perception of preventive mask-wearing, we found it important to investigate if the variables of education, occupation, and assessment of corona measures had a direct impact and could be put in a scientific context. The research question we seek to answer, therefore, is the following:

What influence does the attitude toward COVID-19 containment measures, the level of education, and occupation have when visually assessing interprofessional teams wearing a preventive FFP2 mask?

## 2. Material and Methods

We obtained approval for the study from the Ethics Committee of Witten Herdecke University (S-304/2020). From March to May 2021, we published an online questionnaire via LimeSurvey on social media platforms including Instagram, Xing, LinkedIn, and Facebook, as well as through insurance company networks and healthcare networks. The questionnaire featured four photos of an interprofessional team in different attire, accompanied by 52 questions; see Appendix A.

Participants in the survey could rate the team’s perceived characteristics and their own behaviors based on the following:Likability;Competence;Trust;Choosing the team as a family physician team;Cooperating in the team.

These ratings used a slider with values ranging from 0 to 100, with endpoints labeled “not at all” and “very much”. The rating scale consists of 20 ratings (five/photo) with an excellent internal consistency (Cronbach’s alpha = 0.974).

Socio-demographic data were gathered using questions about the following:Age;Gender;Nationality;Highest educational qualification;Occupation.

Occupational areas, as indicated by the participants, were categorized according to the Classification of Occupations of the Federal Employment Agency, Germany, as outlined in the revised 2020 online edition [39].

Given that the survey took place during the COVID-19 pandemic, additional questions were included regarding the following:Impairment due to the pandemic, rated on a Likert scale from 1 to 6 (ranging from “not at all” to “very strongly”);Assessment of pandemic containment measures in their own country, rated on a Likert scale from 1 to 9 (from “much too lax” through “exactly right” to “completely exaggerated”).

### Statistics

Data were analyzed using IBM SPSS Statistics (version 28). Initial analysis included mean, standard deviation, minimum, and maximum values. The ratings for the four photographs—concerning likability, competence, trust, practice seeking, and cooperation—as well as socio-demographic variables like education, occupation, and assessment of pandemic measures, were evaluated for statistical significance (*p* < 0.05) using MANOVA. Differences in socio-demographic variables underwent post hoc analysis with the Bonferroni correction.

## 3. Results

The questionnaire was accessed 1435 times. Of these, 935 participants completed the questionnaire in its entirety. Upon applying the inclusion criterion of place of residence (Germany, Austria, Switzerland), 906 questionnaires were considered for statistical evaluation.

### 3.1. Sample

Table 1 documents the characteristics of the 906 participants, detailing gender, age groups, place of residence, highest level of education, occupational area, and attitudes toward corona measures. The classification of occupations was based on the specifications provided by the German Federal Employment Agency [39].

### 3.2. Attitude Towards Corona Measures

On a scale of 0–100 (ranging from “too lax” to “too exaggerated”), the average value for attitudes towards the corona measures was 38.15 ± 22.38, indicating a tendency towards viewing the measures as too lax. Comparing groups based on educational attainment, those without a college degree did not differ from those with a degree in their assessment of the corona measures. However, when evaluating based on occupational groups, significant differences emerged. Within the health occupation group, notably fewer participants (N = 77, or 25.7% of that occupational group) rated the corona measures as “too lax” (Chi^2^ = 10.10, df = 4, *p* < 0.039) (refer to Table 1).

### 3.3. Overall Photo Assessment

Across the entire sample (N = 906), the evaluation of the four photos—showing the tunics, coats, masks, and pink socks—revealed the highest ratings for competence, trust, and preference for choosing a practice (see Table 2). Ratings for confidence and the likelihood of choosing a practice were similar across the photos. Teamwork received the lowest scores for all photos, whereas photos showing the pink socks had the highest likability scores. The tunics consistently received lower ratings compared to the scrubs and mask. The pink socks, while associated less with competence, had stronger associations with likability and teamwork (refer to Figure 1).

### 3.4. Education, Profession, and Corona Measures

In the assessment of the four photos with different clothing of the practice team (tunic, coat, mask, pink socks), differences between the group without a university degree (N = 495) versus the group with a university degree (N = 411) became apparent. Regardless of attire, participants without a college degree rated all photos with a higher mean score for likability, competence, confidence, picking a practice, and teamwork than participants with a college degree (see Table 2). The one-factor MANOVA showed no statistically significant difference between the two groups. Post hoc one-factorial ANOVA was performed for each photograph, revealing significant differences between the two groups (all *p* < 0.050) with small effect sizes for liking (all η^2^ < 0.025), competence (all η^2^ < 0.030), confidence (all η^2^ < 0.025), choosing a practice (all η^2^ < 0.025), and team collaboration (all η^2^ < 0.015). The photos with coats (by likability and competence) and with masks (by competence and picking a practice) had the largest difference in ratings by the two educational groups.

The assessment of the four photos with the tunic, coat, mask, and pink socks by the three groups on corona measures showed that the lowest values, and therefore the least favorable assessment, was made by the group “too exaggerated” (N = 306). This group rated almost all photos with a lower average score for sympathy, competence, trust, choosing a practice, and teamwork than participants in the other two groups, “too lax” (N = 294) and “average” (N = 306), which gave similar ratings (see Table 3). Testing with a one-factor MANOVA revealed a statistically significant difference with a small effect size for combined liking (F = 2.92, *p* < 0.008, η^2^ = 0.010, Wilk’s Λ = 0.981), competence (F = 2.53, *p* < 0.019, η^2^ = 0.008, Wilk’s Λ = 0.983), and confidence (F = 2.98, *p* < 0.007, η^2^ = 0.010, Wilk’s Λ = 0.980), but not for choosing a practice (F = 1.16, *p* < 0.323, η2 = 0.004, Wilk’s Λ = 0.992) and team collaboration (F = 0.96, *p* < 0.454, η^2^ = 0.003, Wilk’s Λ = 0.994). Post hoc tests (Bonferroni) showed that mean differences between the three groups on corona measures/sample were not significant (all *p* > 0.050). The largest differences were only seen in the photos with masks. The “too exaggerated” group (N = 306) rated them least favorably for likability, competence, trust, and teamwork; the other two groups rated them comparably better. Only when choosing a practice do the groups “too exaggerated” (N = 306) and “too lax” (N = 294) come to a similar result when assessing the mask.

### 3.5. Corona Measures/Health Professionals

When the four photos were rated based on likability, competence, confidence, choice of practice, and teamwork, using different attire for the practice team, participants from the health profession (N = 300) consistently rated the photos displaying a tunic, coat, mask, and pink socks as less likable compared to the corona Measures group (N = 906) (refer to Table 4). This was consistent regardless of the attire or their attitudes towards corona measures categorized as “too lax” (N = 77), “average” (N = 111), and “too excessive” (N = 112).

A single-factor MANOVA indicated a statistically significant difference with a small effect size among the three groups in terms of overall liking (F = 4.39, *p* < 0.001, η^2^ = 0.043, Wilk’s Λ = 0.916). However, post hoc tests (Bonferroni) found no significant mean differences among the three groups (all *p* > 0.050). The most notable difference was observed in the assessment of the photo with the tunic (refer to Table 3). Interestingly, the photo with the pink socks consistently received the highest likability scores, irrespective of the corona group (refer to Figure 2).

In terms of competence, a single-factor MANOVA showed a statistically significant difference with a small effect size among the three groups (F = 1.76, *p* < 0.106, η^2^ = 0.018, Wilk’s Λ = 0.965). Yet, post hoc tests (Bonferroni) revealed no significant mean differences among the groups (all *p* > 0.050). Notably, the “average” group (N = 306) rated all photos as slightly more competent compared to the other two groups, with the mask photo being perceived as the most competent (see Table 3). The most significant variations among the groups were observed in the assessments of photos with a tunic and pink socks. Both photos received the lowest scores from the “too lax” group (N = 77) (Figure 2).

For confidence, a single-factor MANOVA revealed a statistically significant difference with a small effect size among the three groups (F = 1.88, *p* < 0.083, η^2^ = 0.019, Wilk’s Λ = 0.963). Post hoc tests (Bonferroni), however, showed no significant mean differences among the groups (all *p* > 0.050). The most prominent difference was in the assessment of the pink socks photo. The “too excessive” group (N = 112) gave the pink socks their highest confidence rating (see Table 3). Notably, the mask photo consistently received the highest trust scores across all groups (see Figure 2). The “too lax” group (N = 77) was an exception, giving lower ratings for the tunic, coat, and pink socks compared to the other two groups.

In terms of choosing a practice, a single-factor MANOVA marginally missed statistical significance when comparing the three corona measures/health groups (F = 2.05, *p* < 0.059, η^2^ = 0.020, Wilk’s Λ = 0.960). Post hoc tests (Bonferroni) found no significant mean differences among the three groups (all *p* > 0.050). The most pronounced variations among groups were seen in the assessments of the tunic and pink socks photos. The “too lax” group (N = 77) consistently provided the lowest ratings (see Table 3). In contrast, the “too excessive” group (N = 112) rated the tunic and pink socks photos the highest. Interestingly, the mask photo was rated highest by both the “too lax” (N = 77) and “average” (N = 111) groups (see Figure 2).

Finally, for team collaboration, a single-factor MANOVA showed no statistically significant differences among the three groups (F = 1.15, *p* < 0.333, η^2^ = 0.012, Wilk’s Λ = 0.977). Post hoc tests (Bonferroni) confirmed no significant mean differences among the groups (all *p* > 0.050). The “average” group (N = 306) generally provided slightly more favorable ratings compared to the other two groups for all attire options. However, the “too lax” group (N = 77) consistently rated all photos lower (see Table 3). The only exception was the mask photo, where both the “too lax” (N = 77) and “too excessive” (N = 112) groups gave similar ratings (Figure 2). This trend was previously observed in the corona measures group (see Table 2).

## 4. Discussion

In our initial study on the influence of attire on perceptions of interprofessional teams, participants were shown four photographs, each depicting a team dressed differently [38]. We demonstrated significant differences in the evaluation of interprofessional teams based on clothing. Conducted during the corona pandemic, one of the photographs displayed the team wearing FFP2 masks. This image received notably higher ratings across all age groups for the core values of trust and competence. We understood this heightened perception as being attributable to the prevailing pandemic, with the masked image making a pronounced impact [38].

Prompted by this discovery, we aimed to further understand the visual recognition of preventive masks in healthcare teams. With our 2021 online study giving us an abundance of data, we decided that the socio-demographic variables of education, occupation, and attitude towards COVID-19 containment measures needed further exploration. The occupational category was subdivided into “healthcare”, “education and social services”, and “other”. “Education” was bifurcated into those with and without a university degree. Participants graded their attitude towards COVID-19 containment measures using a Likert scale, ranging from 1 to 9 “too lax (1)” to “average (5)”, to “too exaggerated (9).”

### 4.1. Education

In the educational field, distinct differences emerged regarding the appraisal of professional attire. Participants without a university degree consistently rated all photos higher across parameters like sympathy, competence, trust, choosing a practice, and desire for team collaboration. This suggests that non-degree holders resonated more with the depicted teams than their degree-holding counterparts. We interpret this trend as indicative of a heightened critical attitude among university graduates [16,32].

### 4.2. Occupation

Overall, participants from the health sector evaluated all tested variables far more critical. This could be attributed to the typically higher educational qualifications in the health domain. Higher qualification can be correlated with increased critical competence. Evaluating the occupational field, we see the education sector placing the most trust in preventive masking. Interestingly the health sector exhibited significantly diminished trust in this regard. Employees in the health domain were markedly less inclined to opt for the mask-wearing team. We suggest that healthcare workers might harbor more skepticism about masks than other professionals. There is probably a multitude of reasons for this. The mask might elicit a stronger trigger reaction within their professional milieu. Influenced by the stricter behavioral guidelines and a perceived compulsion to wear masks at all times, a feeling of internal resistance and escalating psychological reactance might be induced within healthcare workers. Feelings of control deprivation are an imminent result [40,41,42]. Stronger emotional responses from mask opponents within the healthcare profession might be explained by this very dynamic.

### 4.3. Attitude Towards COVID-19 Restrictions

It was also important to understand participants’ views towards corona measures in relation to the socio-demographic variables of occupation and education, in helping to gain a deeper understanding in identifying mask opponents. Those rating the corona restrictions “overly strict” consistently rated the core values of likeability, competence, and trust lower for all photographs, especially in relation to the mask image. If we define this subgroup as mask opponents, the lower ratings of the mask photograph become comprehensible [24,43].

### 4.4. Healthcare Sector

Participants from the healthcare sector generally perceived the teams in the photos as less likable, competent, and trustworthy in contrast to the remainder of the study population. Choosing a practice and teamwork were also viewed less favorably. Unraveling the reasons for this is challenging. Within the Health Belief Model’s context, however, such behaviors must bear an inner logic resonating with the individuals’ core beliefs. Correlation of the Health Belief Model (HBM) to its defining variables sees the self-rated assessment of COVID-19 containment measures as the perceived susceptibility/severity. The perceived benefits can be reflected in the attitude towards mask-wearing. Modifying factors are linked to education, occupation, and demographic background.

Habituation to continuous pandemic exposure might induce a casual attitude and fatigue related to mask usage [44,45,46]. An Israeli study highlighted elevated risk behaviors in populations subjected to persistent existential threats [47]. Participants from the education sector, even those skeptical about corona measures, rated the team photos higher for likeability and trust. This phenomenon echoes the observed effects in the non-degree holder evaluation segment. To comprehend the healthcare stakeholders’ behaviors considering their potential as role models, we scrutinized perceptions of corona measures (Likert scale 1–9) from “too lax (1)”, through “average (5)”, and to “too exaggerated (9)”. Remarkably, over a third of healthcare participants (N = 112) who viewed measures as excessive expressed lower concurrence with the core values when visually judging the mask-wearing photo. This was echoed in a 2021 Greek survey where only 71.1% of the polled healthcare professionals intended to receive the corona vaccine. About a third declined vaccination [48]. This observed behavior among healthcare professionals aligns with our data, suggesting that mask opponents might also harbor strong feelings towards corona vaccination. It is disconcerting that over a third of healthcare professionals seem ambivalent towards mask-wearing. Our collective efforts to communicate the substantiated benefits of masks to this pivotal group appear lacking. This mirrors the outcome of an Ethiopian study, where only 33% of 408 participants valued protective masks [49]. Given their frequent interactions with potential COVID-19 patients, healthcare professionals should inherently exhibit robust alignment and competence with mask usage [50]. However, a 2020 Pakistani survey of 392 healthcare participants revealed that 25% displayed inadequate attitudes, knowledge, and mask utilization [51]. An Irish survey from 2021 corroborated this [52]. Our findings aptly fit within this international context. Projecting this deficiency onto the Health Belief Model, the documented evidence advocating mask-wearing, juxtaposed with the perceived mask disadvantages, could cause over a third of healthcare workers to negatively appraise the cost–benefit dynamics. The discomfort associated with mask-wearing, as reported by Berlin hospital staff, might encompass symptoms like breathlessness, cognitive challenges, and fatigue [53]. Intriguingly, the group dynamics across the reviewed studies seem to adhere to the “rule of thirds,” a medical principle dating back to Hippocrates [54,55,56]. This distribution pattern is often seen in behavioral medicine. It should be interpreted with caution. In a survey involving hospital and ambulance staff in South Carolina, a third reported diminished motivation concerning corona protection, resulting in a vaccination rate of only 66% [57]. Behavioral therapy underscores psychological needs as essential behavioral drivers [58]. Ultimately, it is not objective knowledge but subjective attitudes that generate our actions, encapsulated by the saying, “against our better judgment”. Interventions should aim to modulate and cultivate a work environment that bolsters our employees’ health beliefs. A Chinese survey analyzing 273 healthcare worker questionnaires resonated with this sentiment [59].

The study’s contorted female participant ratio might have influenced the outcomes, a limitation previously highlighted. Healthcare sector participants predominantly possessed advanced educational qualifications, potentially biasing the team photo evaluations. Distributing our questionnaire through specific channels might not representatively capture the societal nuances of Germany, Austria, and Switzerland, and thus create a selection bias. Further studies should not only be transmitted through digital channels. Categorizing the “overly stringent” measure skeptics with a scale value of 47–100 might have amplified this group’s representation.

Regarding the study design, it would have been better if the study used a random presentation, giving each participant only one randomly chosen photo for evaluation. Furthermore, if all four photos were presented then at least a random sequence should have been used to average out possible location effects. All participants rating all four photos might introduce a transfer bias and spill-over effects from the photos already assessed.

## 5. Conclusions

Focused research is justified to substantiate mask efficacy. It is crucial to discern if the “mask skeptic” cohort among healthcare workers also resists mandatory vaccination or other preventive actions. Post-pandemic mask-wearing attitudes and motivations need further exploration. Studies pinpointing health behavior motivations will better position us for potential future pandemics. Additional research into interprofessional team perceptions related to mask-wearing, especially among global healthcare workers, is crucial. Such endeavors would further contextualize our study’s unique data. Significant differences were observed in the assessment of practice attire among study participants. Individuals with university degrees exhibited a more critical stance when evaluating practice teams. Profession was identified as a significant variable in the visual assessment of interprofessional teams, with the most contrasting evaluations emerging among participants who perceived the COVID-19 measures as overly stringent. This segment exhibited pronounced reservations when the practice team was portrayed wearing masks. A focal point of our study was to understand participants’ perceptions of the mask-wearing image. Remarkably, an estimated one-third of respondents from the healthcare sector appeared to oppose mask-wearing. This finding warrants further exploration. Within the context of the Health Belief Model, we hypothesize that these individuals, despite their awareness, act in contradiction to prevalent recommendations. This behavior likely aligns with their own personal value systems, suggesting perceived benefits in their choices. It is imperative to engage and sensitize this demographic group, encouraging them to serve as positive role models. To underscore this, a British survey highlighted that participants were primarily motivated to wear masks by a sense of communal welfare [60,61]. Last, but not least, there is the question of how the COVID-19 pandemic will be remembered and what narrative will be passed on to human history. A newly published article examining four studies related to this issue found that memory formation was highly affected by bias and memory distortion. Altogether, there were high behavioral costs for every individual regardless of how anyone dealt with the pandemic [62]. A potentially biased and distorted personal pandemic narrative could again be explained by applying the dynamics of the Health Belief Model [20]. We suggest that the same beliefs we found in one third of healthcare workers regarding mask-wearing may form a biased COVID-19 narrative in the future. For the future, targeted communication and behavioral science-based training programs are needed to strengthen the self-awareness of healthcare workers, thus helping them to understand their importance as role models in future pandemics.

## Figures and Tables

**Figure 1 ijerph-22-01755-f001:**
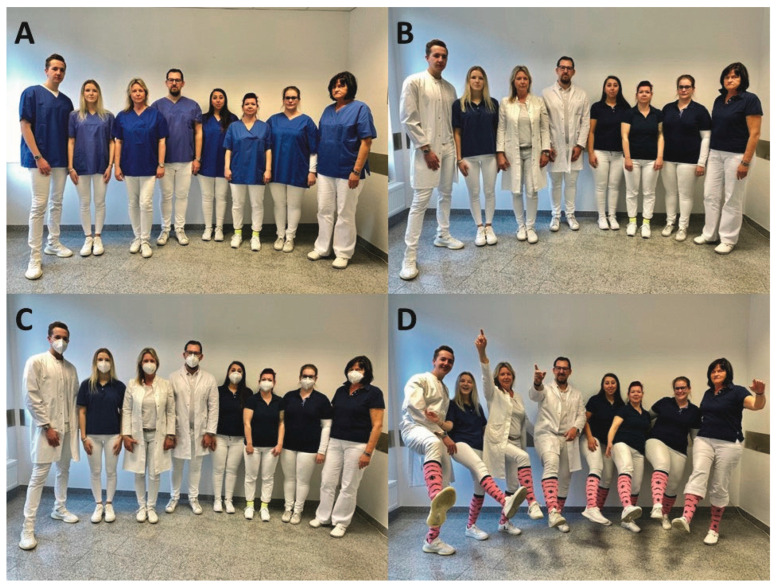
(**A**): Photo featuring an interprofessional team dressed in blue tunics and white pants. (**B**): Photo depicting all team members in white pants and blue polo shirts, with medical team members specifically in white coat (**C**): Photo displaying the team as in Photo **B**, but with all members wearing FFP2 masks. (**D**): Photo showing all team members wearing pink socks (https://pinksocks.life/, accessed on 4 September 2022), appearing in a relaxed mood.

**Figure 2 ijerph-22-01755-f002:**
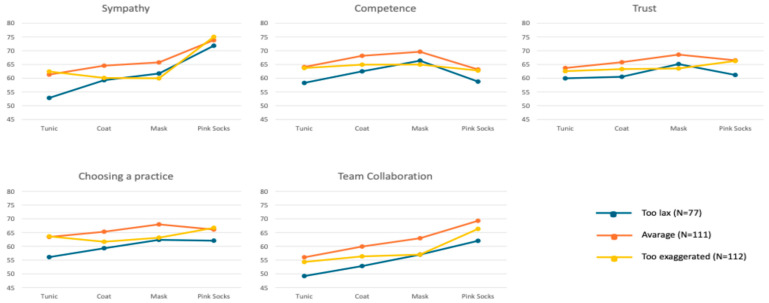
“Health professionals” evaluation of the four photos concerning likability, competence, trust, choosing a practice, and team cooperation by attitude towards governmental measures (mean, standard deviation).

**Table 1 ijerph-22-01755-t001:** Description of study sample (frequency, percent, *n* = 906).

	Variable	#% ^(1)^
Gender	Women	642 (70.86%)
	Men	264 (29.14%)
	Divers	0
Age	18–27 years	205 (22.63%)
	28–37 years	197 (21.74%)
	38–47 years	175 (19.32%)
	48–57 years	193 (21.30%)
	>57 years	136 (15.01%)
Residence	(rather) urban	652 (71.96%)
	(rather) rural	254 (28.06%)
Education	Without university degree	495 (54.64%)
	With university degree	411 (45.36%)
Occupation	Health	300 (33.11%)
	Education and social	152 (16.78%)
	Other	454 (50.11%)
Corona measures	Too lax	294 (32.45%) ^(2)^
	Average	306 (33.77%)
	Exaggerated	306 (33.77%)

^(1)^ All figures in percent refer to the sample (*n* = 906). ^(2)^ Deviations from 100% are due to rounding.

**Table 2 ijerph-22-01755-t002:** Assessment of the four photos by the sample (N = 906) according to likability, competence, trust, choosing a practice, and team cooperation (mean, standard deviation).

	Tunic M ± SD	Coat M ± SD	Mask M ± SD	Pink Socks M ± SD
Sympathy	63.65 ± 24.13	65.58 ± 22.26	65.84 ± 22.18	**76.10 ± 23.35**
Competence	66.93 ± 21.36	69.29 ± 20.40	**70.31 ± 20.63**	65.51 ± 23.95
Trust	66.40 ± 21.86	68.00 ± 21.72	**69.49 ± 21.43**	67.95 ± 23.80
Pick Practice	66.03 ± 25.10	66.92 ± 23.55	**68.58 ± 22.97**	68.40 ± 25.71
Team Collaboration	56.02 ± 25.02	59.13 ± 24.59	60.91 ± 24.01	**67.75 ± 26.94**

**Table 3 ijerph-22-01755-t003:** Assessment of the four photos by the educational groups, professional groups, and groups on corona measures of likability, competence, and trust (mean, standard deviation).

	Tunic M ± SD	Coat M ± SD	Mask M ± SD	Pink Socks M ± SD
**Sympathy**				
**Education Groups**				
Without university degree (N = 495)	66.71 ± 24.74	68.76 ± 22.73	68.64 ± 22.91	79.05 ± 22.11
With university degree (N = 411)	59.98 ± 22.87	61.76 ± 21.08	62.47 ± 20.80	72.54 ± 24.31
**Professional Groups**				
Healthcare (N = 300)	59.56 ± 23.94	61.55 ± 21.01	62.55 ± 20.80	73.78 ± 21.92
Education and social affairs (N = 152)	61.30 ± 22.16	62.35 ± 20.56	64.38 ± 20.25	74.26 ± 23.94
Other (N = 454)	67.15 ± 24.41	69.33 ± 23.01	68.51 ± 23.36	78.24 ± 23.91
**Corona Measures**				
Too lax (N = 294)	62.79 ± 25.27	65.97 ± 23.11	67.57 ± 22.79	76.18 ± 25.06
Average (N = 306)	64.03 ± 23.03	65.67 ± 21.17	67.02 ± 20.38	76.36 ± 22.06
Too exaggerated (N = 306)	64.11 ± 24.14	65.13 ± 22.57	63.01 ± 23.10	75.75 ± 22.98
**Competence**				
**Education Groups**				
Without university degree (N = 495)	69.86 ± 21.83	72.20 ± 21.14	73.43 ± 20.53	67.48 ± 24.17
With university degree (N = 411)	63.41 ± 20.24	65.80 ± 18.91	66.55 ± 20.14	63.13 ± 23.50
**Professional Groups**				
Healthcare (N = 300)	62.46 ± 20.57	65.49 ± 19.24	67.04 ± 19.85	61.91 ± 21.59
Education and social affairs (N = 152)	65.26 ± 19.89	66.36 ± 19.53	69.01 ± 19.20	62.69 ± 23.42
Other (N = 454)	70.45 ± 21.77	72.79 ± 20.86	72.90 ± 21.29	68.83 ± 25.16
**Corona Measures**				
Too lax (N = 294)	67.15 ± 21.87	69.54 ± 21.29	72.01 ± 21.33	67.04 ± 24.27
Average (N = 306)	66.89 ± 20.76	69.67 ± 19.56	71.15 ± 19.56	65.30 ± 23.31
Too exaggerated (N = 306)	66.77 ± 21.52	68.68 ± 20.40	67.83 ± 20.83	64.25 ± 24.27
**Trust**				
**Education groups**				
Without university degree (N = 495)	69.07 ± 22.48	70.85 ± 22.29	72.16 ± 21.69	70.50 ± 23.68
With university degree (N = 411)	63.18 ± 20.66	64.57 ± 20.53	66.28 ± 20.68	64.88 ± 23.61
**Professional Groups**				
Healthcare (N = 300)	62.32 ± 20.80	63.52 ± 20.15	65.81 ± 20.02	65.05 ± 21.72
Education and social affairs (N = 152)	65.19 ± 20.55	65.45 ± 20.93	67.07 ± 20.65	66.07 ± 23.78
Other (N = 454)	69.50 ± 22.52	71.81 ± 22.34	72.74 ± 22.12	70.50 ± 24.86
**Corona Measures**				
Too lax (N = 294)	67.32 ± 22.02	67.98 ± 22.39	71.45 ± 21.72	68.66 ± 24.54
Average (N = 306)	66.09 ± 21.10	67.98 ± 21.00	70.07 ± 20.06	67.86 ± 23.00
Too exaggerated (N = 306)	65.83 ± 22.48	68.04 ± 21.87	67.04 ± 22.29	67.36 ± 23.92

**Table 4 ijerph-22-01755-t004:** “Health professionals” evaluation of the four photos concerning likability, competence, trust, choosing a practice, and team cooperation by attitude towards governmental measures (mean, standard deviation).

	Tunic M ± SD	Coat M ± SD	Mask M ± SD	Pink Socks M ± SD
**Sympathy**				
Too lax (N = 77)	52.84 ± 23.06	59.29 ± 19.86	61.69 ± 20.04	71.83 ± 22.51
Average (N = 111)	61.32 ± 23.56	64.57 ± 20.26	65.76 ± 18.53	73.84 ± 22.44
Too exaggerated (N = 112)	62.43 ± 24.22	60.11 ± 22.29	59.96 ± 23.08	75.06 ± 21.07
**Competence**				
Too lax (N = 77)	58.30 ± 20.14	62.51 ± 18.28	66.36 ± 19.29	58.78 ± 20.27
Average (N = 111)	64.08 ± 20.19	68.13 ± 19.20	69.61 ± 18.94	63.18 ± 22.12
Too exaggerated (N = 112)	63.71 ± 21.01	64.92 ± 19.73	64.96 ± 20.97	62.79 ± 21.90
**Trust**				
Too lax (N = 77)	59.96 ± 19.17	60.51 ± 18.74	65.14 ± 19.82	61.18 ± 20.99
Average (N = 111)	63.71 ± 20.71	65.82 ± 20.46	68.58 ± 19.25	66.50 ± 21.39
Too exaggerated (N = 112)	62.57 ± 21.97	63.32 ± 20.65	63.54 ± 20.73	66.27 ± 22.40
**Choosing a Practice**				
Too lax (N = 77)	56.08 ± 23.62	59.29 ± 21.40	62.35 ± 20.83	62.05 ± 23.13
Average (N = 111)	63.40 ± 24.51	65.29 ± 21.83	67.98 ± 21.12	66.08 ± 25.05
Too exaggerated (N = 112)	63.61 ± 25.01	61.66 ± 22.62	63.14 ± 22.78	66.71 ± 23.85
**Team Collaboration**				
Too lax (N = 77)	49.18 ± 24.23	52.81 ± 23.06	57.00 ± 23.68	61.99 ± 26.70
Average (N = 111)	55.99 ± 23.04	59.90 ± 22.51	62.95 ± 21.09	69.27 ± 23.81
Too exaggerated (N = 112)	54.31 ± 25.09	56.32 ± 22.69	57.00 ± 23.13	66.32 ± 25.81

## Data Availability

The original contributions presented in this study are included in the article/Appendix A. Further inquiries can be directed to the corresponding author(s).

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
