# Peer review of "How Healthcare Teams Appear to the Observer—An Approach to the Complexities of Mask-Wearing Attitudes During the COVID-19 Pandemic"

_ijerph, 2025, doi:10.3390/ijerph22111755_

Round 1
Reviewer 1 Report
Comments and Suggestions for Authors
I have finished the review of the manuscript entitled "“Visual Assessment of Healthcare Teams – An Approach to the Complexities of Mask Wearing Attitudes During the COVID-19 Pandemic, it is a novel study on the pandemic effects with a theoretical orientation from the Health belief model.
Theme falls in the scope of IJERPH and international data are of value.
Nevertheless, certain aspects require further clarification. Is this primarily a re-analysis of previously collected data or an independent study?
Health Belief Model is appropriate and well justified, but underdeveloped. The authors should explicitly map the HBM constructs to their survey items or observed variables.
Classification of "with or without university" sounds weird, I believe you mean with or without "university degree"
The reserach gap should be explicitly presented
Please discuss on potential selection bias and explain measures to reduce it.
Were any pretests or internal consistency analyses (Cronbach’s α) performed
The attitudes toward COVID-19 containment measures ib 1–9 Likert scale would need to be clearly justified. What anchors defined “too lax” vs. “too exaggerated”?
ANOVAs inflates the risk of Type I error, please explain any adjustments or corrections along with effect measures
Tables are too dense, numbers are repeated in tables and text in some cases more than twice.
In discussion, many assertions are speculative and require valid, rogorous references and moderation, such as in the case of "rule of thirds"
Conclussions require rephrasing for mentioning the practical application
Aditionally,
Abstract has too many technical/metholofical details
Please simplify tables, reduce as possible details
Author Response
Dear reviewer 1,
thank you for your thoughtful assessment of our manuscript. All changes and comments we made a part of the uploaded document.
Kind regards HJ Roehrens

Reviewer 2 Report
Comments and Suggestions for Authors
The MS “Visual Assessment of Healthcare Teams – An Approach to the Complexities of Mask Wearing Attitudes During the Covid-19 Pandemic” submitted by Roehrens et al. is an interesting contribution to the debate about attitudes towards mask wearing. The method to assess such attitudes is ingenious as the topic is addressed indirectly by assessing different photos and not asking about masks at all. The paper suffers however from a number of shortcomings as detailed below.
General remark:
Do not use the jargon terms ‘corona pandemic’ or ‘corona measures’ etc. Whenever you refer to COVID-19 state it explicitly.
Specific remarks:
- Title: The title is possibly misleading and I suggest to change it as follows: “How Healthcare Teams Appear to the Observer – An Approach to the Complexities of Mask Wearing Attitudes During the Covid-19 Pandemic”
- Abstract: Line 10: Omit the comma after ‘explores’; ‘attitude towards…’ is not a sociodemographic factor! Line 16: replace the comma after ‘education’ by and ‘and’. Line 18: What three groups? Nothing about groups has been said before, omit it. The sentence ‘Key finding…picture’ should be deleted. Simply state what the findings were. Line 25: Replace ‘10’ by ‘19’.
- Line 35/36: Write: ‘This topic was central in the debate about governmental protective measures all over the world’.
- Line 37: Replace ‘corona’ by ‘SARS-CoV-2’
- Lines 67/68: Replace by: “This will likely be relevant also in future pandemics’
- Line 69: Place a ‘ after ‘individuals’
- Line 84: Write: ‘…examined, prior to this study, the influence…’
- Line 91: Write ‘The study presented here…’
- Study design: In my opinion, the design of the study was not optimal. Apparently all participants rated all 4 photos thus introducing a bias from transfer and spill-over effects from the photos already assessed. It would have been better if the study used a random presentation giving each participant only one randomly chosen photo for evaluation. Furthermore, if all 4 photos are presented then at least a random sequence should have been used to average out possible location effects. This should be mentioned in the discussion.
- Statistics: The evaluation has been done considering only one of each potential factor separately (education, occupation etc.) and only for health professionals a subgroup comparison was conducted. I suggest to perform a simultaneous analysis (not by ANOVA or MANOVA but by applying a GLM offering greater flexibility) for all assessments including main effects and 2-variable interactions in the model.
- Line 148: Use a comma not a decimal point in ‘1,435’
- Table 2: Write out ‘HSn’
- Line 230: I think the comparison here is wrong. What is ‘Corona Measures/Sample’ anyway?
- Table 2: The header must be changed: ‘Health professionals’ evaluation of the four photos concerning likability, competence, trust, choosing a practice, and team cooperation by attitude towards governmental measures (mean, standard deviation)’
- You can collapse the 5 figures into one with five panels (A to E) in a 2x3 or 3x2 frame. Include an x-axis label (mean score, range 0-100). Change the figure legend in accordance with Table 2.
Author Response
Dear Reviewer 2,
we sincerely thank you for your thoughtful assessment of our manuscript. In our upload we indicate all our changes made.
Kind regards HJ Roehrens

Round 2
Reviewer 2 Report
Comments and Suggestions for Authors
You have satisfactorily addressed my concerns and revised the MS accordingly. Only two things need to be changed but this can be done during editorial processing.
- The two sentences lines 13-14 are unclear and maybe wrong. Instead write: 'In addition, eduacation and occupation of participants were obtained'. Since you did not ask about COVID-19 measures the second sentence is misleading. I suggest: "By measuring the attitudes towards an interprofessional team wearing FFP2 masks implicitly the attitude towards this COVID-19 measure is assessed"
- Line 120: Data is a plural word, hence write 'data were'